# MR Imaging Biomarkers for Clinical Impairment and Disease Progression in Patients with Shoulder Adhesive Capsulitis: A Prospective Study

**DOI:** 10.3390/jcm10173882

**Published:** 2021-08-29

**Authors:** Romain Gillet, François Zhu, Pierre Padoin, Aymeric Rauch, Gabriela Hossu, Pedro Augusto Gondim Teixeira, Alain Blum

**Affiliations:** 1Guilloz Imaging Department, Central Hospital, University Hospital Center of Nancy, 54000 Nancy, France; f.zhu@chru-nancy.fr (F.Z.); pierre.padoin@gmail.com (P.P.); aym.rauch@gmail.com (A.R.); p.teixeira@chru-nancy.fr (P.A.G.T.); a.blum@chru-nancy.fr (A.B.); 2CIC-IT, CHRU Nancy, Université de Lorraine, 54000 Nancy, France; g.hossu@chru-nancy.fr

**Keywords:** adhesive capsulitis, MRI, shoulder, constant-murley score, inferior gleno-humeral ligament

## Abstract

Background: MRI diagnostic criteria of shoulder adhesive capsulitis (AC) are nowadays widely used, but there is little information available on the association between MRI findings and clinical impairment. Purpose: To determine the correlation of MRI findings with the Constant–Murley Score (CMS), pain duration and symptoms at the one-year follow-up in AC patients. Materials and methods: This monocentric prospective study included 132 patients with a clinical diagnosis of shoulder AC who underwent shoulder MRI. Mean patient age was 54.1 ± 9.3 years, and there were 55 men and 77 women. A radiologist examined all patients and completed the CMS just prior to MRI. Pain duration was assessed along with the signal intensity and measured the maximal thickness of the inferior glenohumeral ligament (IGHL) by two radiologists. Medical record analysis was performed in a sub-group of 49 patients to assess prognosis approximately one year after the MRI examination. Linear regression analysis with the Pearson test and the Fisher exact test were used to determine the association between MRI findings and clinical impairment. Results: There was a significant difference in mean pain duration score (3.8 ± 1.2 versus 3.2 ± 0.9 and 3.8 ± 1.2 versus 3.2 ± 0.9, respectively, for readers 1 and 2) and in mean mobility scores (15.7 ± 8 points versus 19.6 ± 10.1 points and 15.8 ± 8.2 points versus 19.4 ± 10 points, respectively, for readers 1 and 2) in patients with a high IGHL signal compared to those with a low IGHL signal (*p* < 0.05). IGHL was thicker in patients with clinical improvement at one-year follow-up compared to those presenting clinical stability or worsening (*p* < 0.05). Conclusions: In patients with shoulder AC, the degree of signal intensity at the IGHL was inversely related to shoulder pain duration and range of motion, and a thickened IGHL indicated a favorable outcome at one-year follow-up.

## 1. Introduction

Adhesive capsulitis (AC) of the shoulder is a common condition with an incidence in the general population varying considerably from 2 to 5.3% for primary and from 4.3 to 38% for secondary AC (e.g., AC preceded by a clinical or surgical event) [1]. Although spontaneous resolution is the rule, years can ensue (mean 18–30 months) before joint mobility returns to normal [2]. Various treatment options exist for AC (e.g., oral anti-inflammatory drugs, intraarticular corticoid injection, physiotherapy, percutaneous capsular distention, surgical release, etc.) depending on the level of clinical impairment, and on an accurate diagnosis. Thus, disease staging and identification of inflammatory changes could have an impact on patient management [3].

AC is classically diagnosed based on clinical presentation, medical history, and physical examination. Diagnosing this condition, however, can be challenging as AC may occur in various clinical scenarios and has multiple potential differential diagnoses (e.g., rotator cuff tears, calcifying tendonitis, osteoarthritis, inflammatory tumors…) [2]. Imaging plays an ever-growing role in the evaluation of patients with suspected AC, ruling out pathologic conditions that can clinically mimic AC [4], and in diagnostic confirmation when clinical findings are equivocal [5,6,7,8,9,10,11,12,13,14]. AC-suggestive MRI findings are well recognized and primarily involve inferior glenohumeral ligament (IGHL) (hypersignal and thickening) and rotator interval (RI) scarring and inflammation [7,9,10,12,15,16,17].

Patients with AC typically complain of a gradual and progressive onset of pain, sleep-disturbing night pain, and active and passive limitation at various degrees of ranges of motion (ROM), both in elevation and rotation, for at least 1 month [18]. The Constant–Murley Score (CMS) is often used to evaluate the impact of AC in shoulder function, with potential implications in patient management [19]. Although the correlation of MRI findings with clinical staging was demonstrated in 2008 by Sofka et al. [20], there is little information available on the association between MRI findings and clinical impairment, which could be important for therapeutic decision making [21,22,23,24,25,26,27]. We hypothesize MR imaging signs, particularly the IGHL signal and thicknesses, could serve as biomarkers for shoulder function impairment and AC progression over time. The aim of our study was to evaluate the correlation between MRI findings in AC patients, CMS, and symptoms at the one-year follow-up. 

## 2. Material and Methods

### 2.1. Study Group

Our institutional review board approved this study, and all patients gave written informed consent. From 10 October 2013 to 16 October 2017, 170 patients over 18 years of age were enrolled prospectively and consecutively. These patients had been referred by orthopedic surgeons or rheumatologists due to the clinical diagnosis of shoulder AC and underwent shoulder radiographs and MRI. 

Patients with MRI contraindications, prior shoulder surgery, severe rotator cuff damage with at least a full-thickness tear of one tendon, shoulder osteoarthritis (osteophytes on radiographs), calcific tendinosis, shoulder bursitis, biceps tendinosis, and fractures on MRI were excluded. One patient withdrew from the study; four were excluded because of missing clinical data, and 33 because of extensive rotator cuff damage. Thus, the final study population consisted of 132 patients with a mean age of 54.1 ± 9.3 (22–78) years. There were 55 men (mean age 53.5 ± 8.8 (22–70) years) and 77 women (mean age 54.4 ± 10.8 (22–78) years) with a M/F sex ratio of 0.63. Two patients were suspected of having bilateral AC, yielding 134 shoulder MRI studies.

### 2.2. Shoulder Function Assessment

A modified CMS was applied to all patients by a senior radiologist just prior to the MRI examination [19]. Two subjective variables for a maximum score of 35 were evaluated: daily living pain (varying from 0—severe pain to 15 points—no pain) and daily living activity limitation (varying from 0—maximal limitation to 20 points—no limitation). The patients answered a questionnaire assessing the degree of pain (no pain, slight, moderate, or severe pain), activity level (pain during work, sports and recreation, sleep) and arm range of motion (ROM) (arm elevation up to the waist, xiphoid process, neck, top of the head, above the head). The examiner received prior training on performing the CMS. ROM was also quantitatively assessed with a goniometer in a seated position, in external and internal rotation, forward and lateral elevation, and scored in each position by the examiner (varying 0–30° = 0 to 151–180° = 10 points for each movement). Thus, the ROM score varied from 0—minimal mobility to 40—maximal mobility). The final CMS, therefore, ranged from 0, indicating a highly impaired shoulder function to 75 points, indicating a normal shoulder function (Appendix A). Shoulder strength, which was part of the original CMS, was not evaluated in this study, because no reliable measurement device was available in our department.

In addition to the modified CMS score, the pain duration was graded as follows: 1 = less than 6 weeks2 = between 6 weeks and 3 months3 = between 3 and 6 months4 = between 6 months and 1 year5 = over 1 year

The presence of diurnal pain, nocturnal pain, and nocturnal pain predominance were also evaluated. 

### 2.3. Clinical Follow-Up

A clinical follow-up was available in a sub-group of 49 patients with a mean age of 54 ± 8.8 (37–74) years treated by physical therapy. Other patients were lost to follow-up or were treated in other institutions. There were 17 men (mean age 50.9 ± 6.6 (38–61) years) and 32 women (mean age 55.7 ± 9.6 (37–74) years) with a M/F sex ratio of 0.53. Based on medical record data (pain, activities, and ROM), and the symptoms at 9 to 13 months after the MRI examination were classified as improved, stable, or worsened. None of these patients had been treated by intra-articular corticosteroid injection.

### 2.4. Mri Examination

MRI examinations were performed with either a 1.5 T (105 patients) or a 3.0 T (27 patients) scanner (Signa HDxt, GE Healthcare, Milwaukee, WI, USA) using a dedicated eight-channel shoulder coil and similar protocols. 

All MRI examinations consisted of an axial and oblique sagittal fast spin-echo (FSE) T1-weighted acquisitions (at 1.5 T: TR/TE, 500/10; echo-train length (ETL), 2; matrix, 352 × 320; NEX, 0.5; FOV, 160 mm; gap, 5 mm; slice thickness, 4 mm; at 3.0 T: TR/TE, 740/minimum full; ETL, 2; matrix, 352 × 256; NEX, 1; FOV, 150 mm; gap, 1 mm; slice thickness, 3 mm); axial, oblique sagittal and oblique coronal FSE T2-weighted fat-saturated images (at 1.5 T: TR/TE, 3500/50; ETL, 12; matrix, 384 × 320; NEX, 1.5; FOV, 160 mm; gap, 3.9 mm; slice thickness, 3.5 mm; at 3.0 T: TR/TE, 3040/45; ETL, 11; matrix, 352 × 256; NEX, 2; FOV, 150 mm; gap, 0.3 mm; slice thickness, 3 mm).

### 2.5. Image Analysis

The images were retrospectively reviewed by two musculoskeletal radiologists with three (FZ) and seven years (PP) of clinical experience with MRI using a PACS station (Synapse^®^, v4.1.600, Fujifilm, Montigny, France). A third radiologist (P.A.G.T.) with 11 years of clinical experience with MRI performed a training session with the two readers with 20 MRI studies of patients with AC, not included in the study population prior to the readouts. The readers were blinded to clinical and demographic data. 

The signal intensity of the IGHL on oblique coronal T2-weighted fat-saturated images was graded as follows (Figure 1):1: normal homogenous low signal intensity2: partial or foci of signal hyperintensity3: global signal hyperintensity4: linear hyperintensity of the peri-articular soft tissues

The whole ligamentous complex was considered in the analysis: the anterior band, posterior band, and hammock portion. The patients with IGHL scores of 1 and 2 were considered to have a low IGHL signal intensity, and those with grades 3 and 4 were considered to have high IGHL signal intensity. The thickness of the IGHL was measured at the glenoidal and humeral insertions on oblique coronal T2-weighted fat-saturated images, according to Mengiardi et al. [9] and classified as <4 mm, between 4 and <6 mm and ≥6 mm (Figure 2) [28]. The thickest portion of the coracohumeral ligament (CHL) was measured on the sagittal T2-weighted fat-saturated images, according to Mengiardi et al. [9] (Figure 3). The size of the axillary recess and superior glenohumeral ligament thickness were not assessed as MR arthrograms were not available. 

### 2.6. Statistical Analysis

The R Development Core Team software (version 3.0.12013, R Foundation for Statistical Computing, Vienna, Austria) was used to perform statistical analysis. Statistical significance for all tests was defined as *p* < 0.05. Quantitative data are presented as mean ± standard deviation (range). 

Linear regression analysis with the Pearson test was used to evaluate the correlation between the signs of AC studied on MRI and pain, mobility, activity scores, and pain duration. The association between MRI findings, global modified CMS score, diurnal pain, night pain, and predominance of night pain was assessed with the Fisher exact test. The association between MRI findings and clinical follow-up was assessed with the Wilcoxon test. For each MRI measurement, intraclass correlation coefficients (ICC) were calculated to assess interobserver variability. ICC values below 0.5 were considered poor, between 0.5 and 0.75 moderate, between 0.75 and 0.90 good, and above 0.9 excellent [29].

## 3. Results

Table 1 shows demographic characteristics and modified CMS in the study population. The mean global modified CMS was 31.3 ± 14.2 (2–69) points, and the mean pain duration grade was 3.5 ± 1.1 (1–5). Clinical pain characteristics are shown in Table 2. Night pain was frequent and predominant in about half of the concerned patients. Table 3 shows the pain duration grade in each grade of IGHL signal intensity. IGHL signal intensity was low in 70 shoulders (52.2%) and high in 64 (47.8%) for reader 1. These figures were 72 (53.7%) and 62 (46.3%), respectively, for reader 2. Table 4 shows the MRI findings in the shoulders evaluated.

ICC was excellent in grading IGHL signal as low or high (0.96), and moderate when taking in account all the four grades (0.67). ICC values were moderate for IGHL thickness (glenoidal insertion: 0.72, humeral insertion: 0.61) and poor for CHL thickness (0.09). 

Mobility scores were significantly different in patients with high IGHL signal intensity compared to those with low intensity for both readers (*p* = 0.04 and 0.02 for readers 1 and 2). The mean mobility scores between shoulders with low and high IGHL signal intensity grades were 19.6 ± 10.1 (2–40) points versus 15.7 ± 8 (0–38) points for readers 1 and 19.4 ± 10 (0–40) points versus 15.8 ± 8.2 (0–38) points for reader 2. The variation of mobility scores with respect to IGHL signal intensity grade is shown in Figure 4. 

For both readers, pain duration was significantly shorter in patients with high IGHL signal intensity compared to those with a low signal IGHL (*p* = 0.03 and 0.04 for readers 1 and 2). The pain duration grades in patients with low and high IGHL signal intensity were 3.8 ± 1.2 (1–5) versus 3.2 ± 0.9 (1–5) for reader 1 and 3.8 ± 1.2 (1–5) versus 3.2 ± 0.9 (1–5) for reader 2. Similarly, as the IGHL signal intensity grade increased, there was also a decrease in mean pain duration for both readers (Table 3 and Figure 5). The presence of high IGHL signal intensity was also significantly associated with nocturnal pain predominance for both readers (*p* = 0.003 and 0.003). 

The glenoidal IGHL thickness was significantly correlated with activity limitation scores for reader 1 (*p* = 0.005). Patients with IGHL measuring < 4 mm, between 4 and <6 mm, and ≥6 mm presented a progressive increase in activity limitation scores of 8.9 ± 5 (0–20) points, 9.7 ± 4.5 (0–20) points, and 11.5 ± 3.8 (4–20) points, respectively for reader 1. For reader 2, these figures were 8.4 ± 3.9 (0–20) points, 10.5 ± 4.7 (2–20) points, and 9.9 ± 4.6 (0–20) points, respectively, which suggest a similar tendency for values < 6 mm, but this variation was not statistically significant (*p* = 0.09). The IGHL thickness at the humeral insertion was significantly associated with pain duration for both readers (*p* = 0.04 and 0.02). For reader 1, with an increasing humeral IGHL thickness, the pain duration decreased (pain duration grades of 3.6 ± 1.1 (1–5), 3.3 ± 1.1 (1–5) and 3.3 ± 1.2 (1–5) points for patients with IGHL thicknesses of <4 mm, between 4 and <6 mm and ≥6 mm, respectively). For reader 2, the same tendency was found for patients with IGHL of <6 mm in thickness; however, for patients with ligaments ≥ 6 mm, the pain duration was longer (3.6 ± 0.8 [3,4,5] points). 

CHL measurements are shown in Table 4. This ligament could not be measured confidently in five patients for reader 1 and 20 patients for reader 2. There was no association between CHL thickness and clinical impairment.

Concerning disease progression, 31 patients (13 men, 18 women, mean age 55 ± 9.1 (38–74) years) showed improvement, 11 patients (2 men, 9 women, mean age 55 ± 10.1 (37–67) years) stability, and 7 worsening (2 men, 5 women, mean age 54 ± 6 (49–67) years). IGHL thickness was significantly correlated with clinical outcomes on the humeral side for both readers (*p* = 0.005 and 0.04 for readers 1 and 2) and on the glenoidal side for reader 1 (*p* = 0.002 and 0.05 for readers 1 and 2). Patients with clinical improvement had thicker IGHL on its humeral side (4 ± 1.5 (2–8) mm and 4 ± 1.3 (2–7) mm for readers 1 and 2) than those with worsening (2.2 ± 0.4 (2,3) mm and 2.7 ± 0.7 (2–4) mm for readers 1 and 2). For reader 1, patients with a stable clinical outcome also had a thicker IGHL than those with worsening, on both sides (glenoidal side: *p* = 0.02, humeral side *p* = 0.005). The same tendency was observed for reader 2, but these differences were not statistically significant (*p* = 0.2). For both readers, the presence of high IGHL signal intensity was not significantly correlated with disease progression (*p* > 0.05). In patients with worsening, IGHL was found to be ≤3 mm in 66 to 83% for reader 1, and 83% to 100% for reader 2. IGHL thickness distribution according to clinical outcomes is shown in Figure 6.

For both readers, there was no association between CMS modified global score, pain intensity grade, diurnal pain, and MRI findings.

## 4. Discussion

Our study showed a significant correlation between high IGHL signal intensity and the pain duration in patients with AC, with a clear high signal predominance in the patients presenting with pain from three to six months. Additionally, the reproducibility for the differentiation between low- and high-signal IGHL was considered excellent. Those results are in agreement with Sofka et al. [20], who stated that capsular high signal intensity in the axillary pouch was most closely associated with pain from three to nine months. High IGHL signal was also associated with night pain, which may have a negative impact on sleep quality and mental health [30,31]. Another important finding was the significant decrease in mobility scores in patients with high IGHL signal hyperintensity, which was not previously reported. Prior reports have also indicated that capsular edema and rotator interval signal abnormalities, were independent predictors of a better outcome for pain relief after glenohumeral corticosteroid injections, confirming the inflammatory nature of these MR findings [32]. In light of these results, increased IGHL signal intensity in patients with AC can be considered as a maker for an early inflammatory disease stage, and is associated with inflammatory-type pain and limited ROM. Thus, the treatment of patients with such finding should be aimed at reducing (e.g., intra-articular corticosteroid injection, cryotherapy) or limiting (e.g., gentle physiotherapy) the capsular inflammatory process [33,34,35,36,37]. Previous studies demonstrated that MRI could not predict AC prognosis [21] or the outcome after capsular distension [38]. Our results, however, indicate that patients with a thick IGHL (4 mm or higher, particularly on the humeral side) were very likely to have a favorable outcome at follow-up (performed approximately one year after imaging). Conversely, thin IGHL (3 mm or lower) was associated with clinical worsening. This could be related to collagen accumulation in the joint capsule in late disease phases [39,40]. As AC is a disease with a self-limited course, a thicker IGHL could be an indicator of a late disease phase and hence be associated with a favorable outcome. Similar to signal changes, the IGHL thickness may have implications in the therapeutic decision-making of AC patients. Although further studies are necessary, these patients might be more suitable for therapeutic options aiming at decreasing capsular stiffness (e.g., hydrodilation, physiotherapy with active-assisted ROM exercises, stretching, and muscle strengthening) [41,42,43]. 

Unlike Anh et al. [26], we did not find any correlation between MRI findings and the degree of pain, but we did not rate IGHL enhancement, as its signal on T2-weighted fat-saturated FSE images has been shown to be reliable without improvement after gadolinium injection [6]. Additionally, systematic gadolinium injection is not currently recommended for the evaluation of patients with AC and should be reserved for patients with equivocal findings on conventional sequences [44,45]. None of the MRI findings evaluated was correlated with global CMS results, in agreement with Park et al. [23]. Capsule thickness in the axillary recess has been described as a reliable diagnostic tool of AC when >4 mm [28,46], but in our study, less than 30% of patients fulfilled this criterion at the humeral insertion. Contrary to the presented results, some authors indicated that capsular thickness on ultrasound and MRI was associated with shoulder function impairment [23,46]. We hypothesize that these differences are related to patient selection bias, and that IGHL thickness could be a more reliable diagnostic tool in patients’ later AC phases, whereas in earlier disease phases the implications of this finding could be less clear. 

This study has limitations. Most importantly, AC diagnosis was confirmed neither by arthroscopy nor histologically. However, clinical findings remain the basis for the diagnosis of AC, and the diagnostic performance of MRI diagnostic criteria has been previously evaluated [6,7,8,10,11,12,17,46,47]. As the estimation of pain duration provided by patients may be imprecise, a pain score system with time intervals was used to limit this potential bias. Since the correlation between MRI findings and the range of motion in each direction is still debated [23,24,25,46], we preferred to analyze global motion scores only, which could be responsible for some of the differences between the presented results and prior reports. There was no control group and no systematic clinical or MRI follow-up of the patients included. We did not rate intra-observer agreement. The study population is relatively heterogeneous with various disease stages and clinical impairment levels; however, the study population is one of the largest reported so far and is representative of routine clinical practice. Finally, the AC etiology (e.g., idiopathic versus secondary) could have an impact on the natural disease course and was not considered in this study. 

In conclusion, two potentially useful MR biomarkers in patients with AC could be identified. First, the increased T2 signal intensity at the IGHL, which is an indicator of an early inflammatory phase AC and is associated with recent pain (3–6 months), nocturnal pain, and decreased ROM. Secondly, the thickness of the IGHL was significantly related to the clinical outcome (>4 mm is associated with a favorable outcome, whereas <3 mm with a worse prognosis). These findings should be considered in the MRI evaluation of patients with AC, with likely therapeutic implications. 

## Figures and Tables

**Figure 1 jcm-10-03882-f001:**
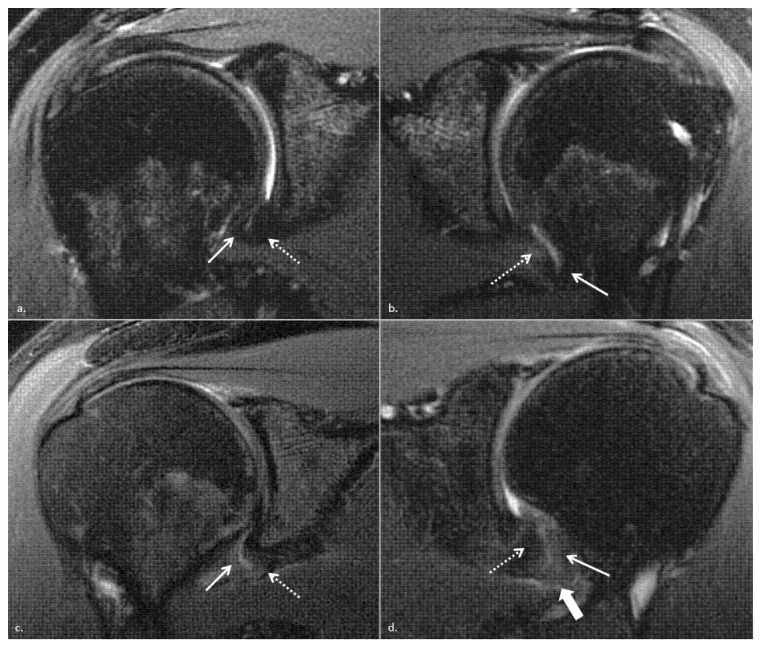
(**a**–**d**) Frontal oblique non-contrast fat-suppressed T2-weighted fast spin-echo MRI shows method used to grade glenohumeral inferior ligament signal on its glenoidal (white arrow) and humeral (dotted arrow) insertions, without signal abnormality in the right shoulder of a 55-year-old woman with adhesive capsulitis graded 1 in (**a**), a discontinuous glenoidal side IGHL hypersignal graded 2 in the left shoulder of the same woman with contralateral adhesive capsulitis in (**b**), a global both side IGHL hypersignal graded 3 in the left shoulder of a 43-year-old man with adhesive capsulitis in (**c**) and an overflow of the hypersignal in adjacent soft tissues (thick white arrow) graded 4 in the right shoulder of a 46 years old man with adhesive capsulitis in (**d**).

**Figure 2 jcm-10-03882-f002:**
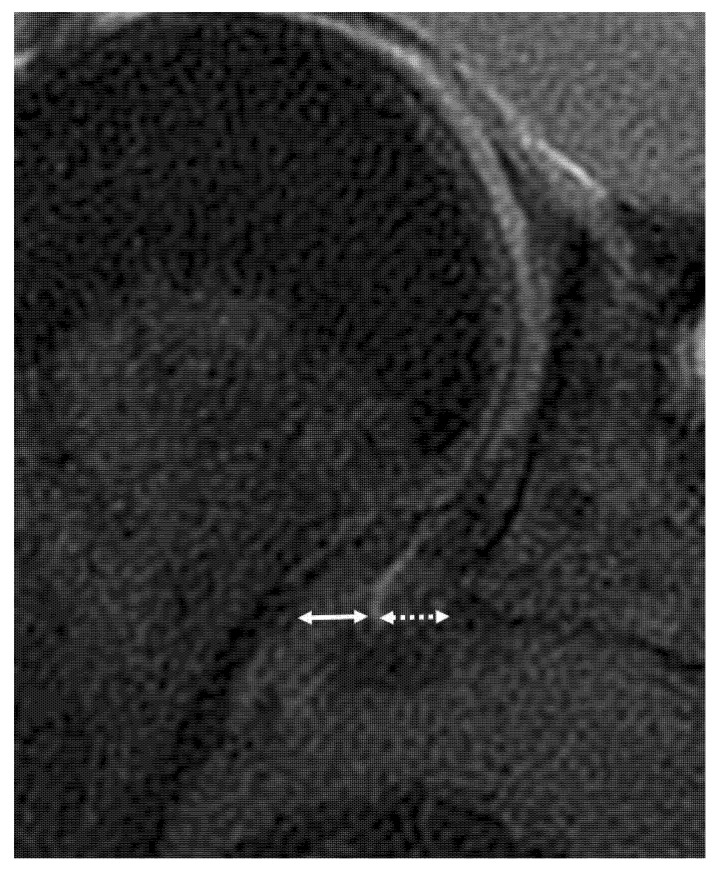
Coronal oblique non-contrast fat-suppressed T2-weighted fast spin-echo MRI of the right shoulder in a 44-year-old woman with adhesive capsulitis shows method used to measure inferior glenohumeral ligament thickness at its glenoidal (dotted double arrow) (4 mm) and humeral insertion (double arrow) (4.5 mm).

**Figure 3 jcm-10-03882-f003:**
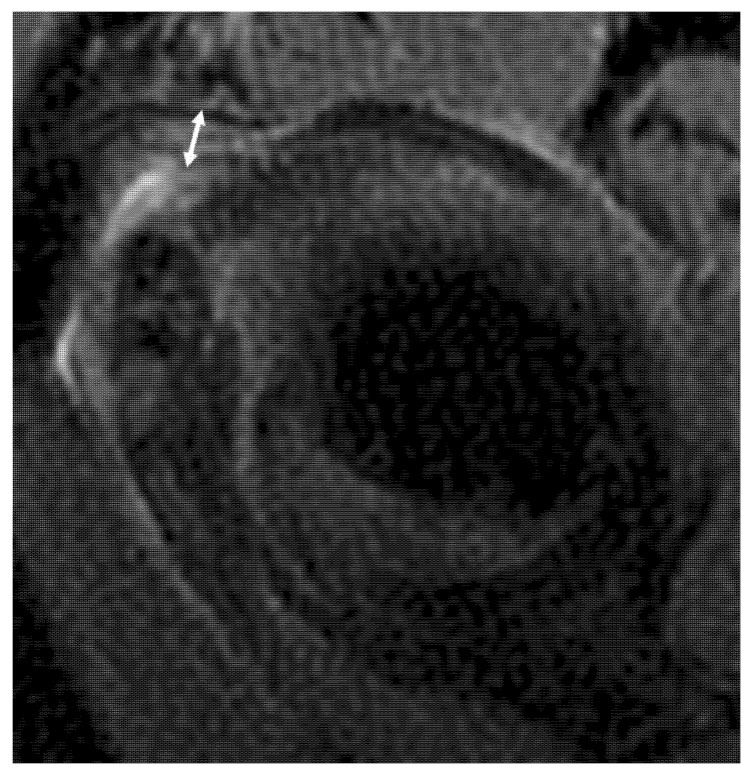
Sagittal oblique non-contrast fat-suppressed T2-weighted fast spin-echo MRI of the left shoulder in a 55-year-old man with adhesive capsulitis shows method used to measure coracohumeral ligament thickness (double white arrow) (4 mm). Additionally, note its high signal intensity.

**Figure 4 jcm-10-03882-f004:**
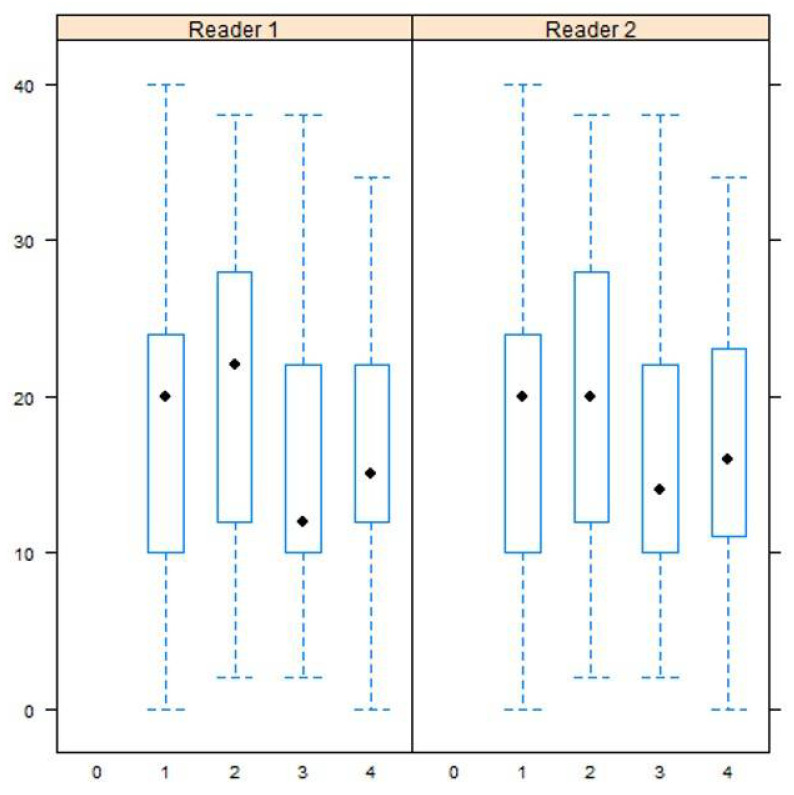
Box-plot showing mean mobility score (*y*-axis) according to inferior glenohumeral ligament intensity grade (*x*-axis) for reader 1 and reader 2.

**Figure 5 jcm-10-03882-f005:**
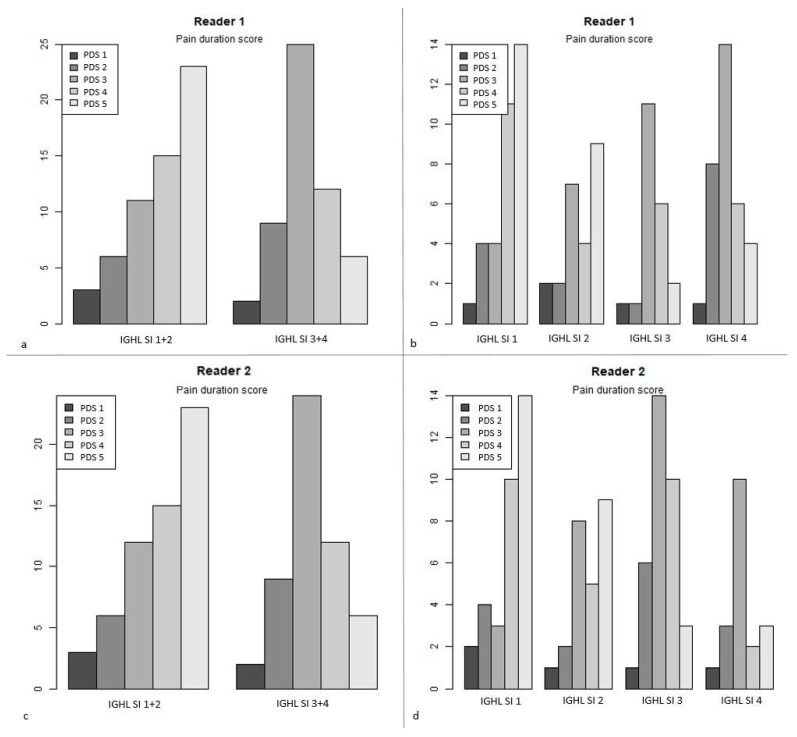
Bar plot representing pain score duration (PDS) (number of patients on *y*-axis) according to inferior glenohumeral ligament signal intensity (IGHL SI) (*x*-axis), shown for reader 1 for grade 1 + 2 and 3 + 4 in (**a**), for each grade (1, 2, 3, 4) in (**b**), and for reader 2 for grade 1 + 2 and 3 + 4 in (**c**), for each grade in (**d**).

**Figure 6 jcm-10-03882-f006:**
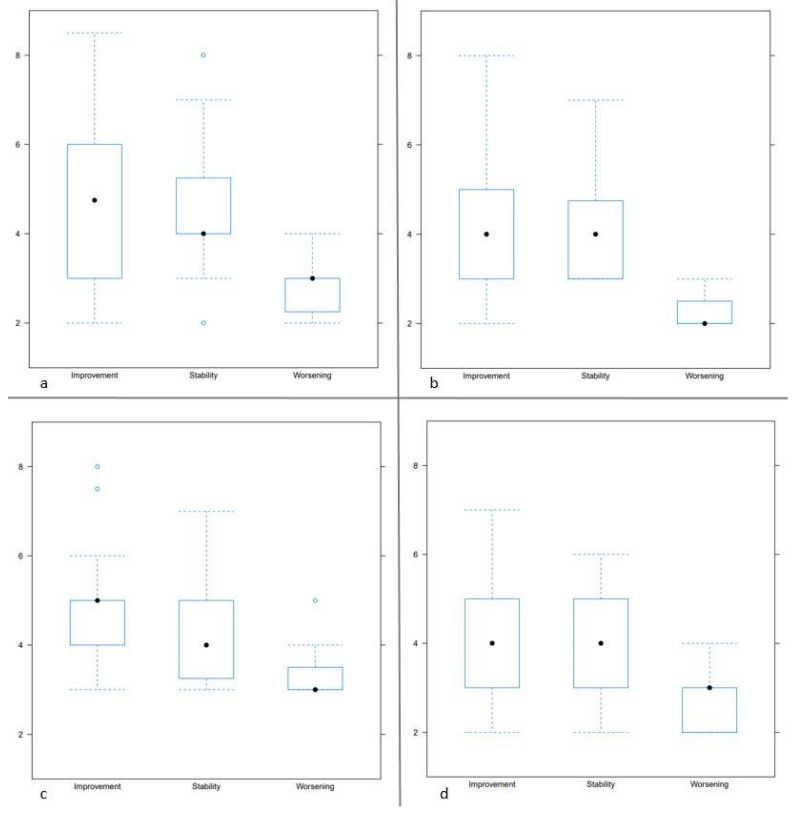
Box plot representing inferior glenohumeral ligament thickness (*y*-axis) according to clinical outcomes for reader 1 on the glenoidal (**a**) and humeral side (**b**), and for reader 2 on the glenoidal (**c**) and humeral side (**d**).

**Table 1 jcm-10-03882-t001:** Summary of patients’ ages and clinical impairment items.

Parameter	Minimum	Maximum	Mean	Standard Deviation
Patient Age				
All patients (*n* = 132)	22	78	54.1	9.3
Men (*n* = 55)	22	70	53.5	8.8
Women (*n* = 77)	22	78	54.4	10.8
Modified Constant-Murray score	2	69	31.3	14.2
Pain Intensity Score	0	15	4	3.8
Activity Score	0	20	9.8	4.5
Mobility Score	0	40	17.7	9.3
Pain Duration Grade	1	5	3.5	1.1

**Table 2 jcm-10-03882-t002:** Summary of patients’ pain characteristics.

Parameter	Effective
Pain Duration Grade	*n* = 112 *
1	4.5% (*n* = 5)
2	13.4% (*n* = 15)
3	32.1% (*n* = 36)
4	24.1% (*n* = 27)
5	25.9% (*n* = 29)
Diurnal Pain	94.7% (*n* = 127)
Night Pain	87.3% (*n* = 117)
Predominance of Night Pain	46.2% (*n* = 62)

* 22 data were missing because patients were not able to determine it.

**Table 3 jcm-10-03882-t003:** Pain duration grade according to inferior glenohumeral ligament signal intensity grade.

IGHL Signal Intensity				Pain Duration Grade	
Reader 1		Reader 2		Reader 1	Reader 2
1	(*n* = 34)	1	(*n* = 34)	3.9 ± 1.1	3.8 ± 1.2
2	(*n* = 24)	2	(*n* = 25)	3.6 ± 1.3	3.7 ± 1.1
3	(*n* = 21)	3	(*n* = 34)	3.3 ± 0.9	3.2 ± 0.9
4	(*n* = 33)	4	(*n* = 19)	3.1 ± 1	3.1 ± 1

IGHL: inferior glenohumeral ligament, results of pain duration score are presented as mean ± standard deviation. Range of all sub-groups of pain duration grade was 1–5.

**Table 4 jcm-10-03882-t004:** Summary of patients’ MRI measurements.

Parameter	Minimum		Maximum		Mean		Standard Deviation	
	R1	R2	R1	R2	R1	R2	R1	R2
IGHL thickness (glenoidal side)	2	2	10	8	4.3	4.5	1.3	1.2
IGHL thickness (humeral side)	2	2	8	7	3.8	3.7	1.3	1.2
CHL thickness	1.5	1	5	4	2.5	2.2	0.6	0.6

Values are given in millimeters. IGHL: inferior glenohumeral ligament, CHL: coracohumeral ligament. R1: Reader 1, R2: Reader 2.

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
