# Peer review of "MR Imaging Biomarkers for Clinical Impairment and Disease Progression in Patients with Shoulder Adhesive Capsulitis: A Prospective Study"

_jcm, 2021, doi:10.3390/jcm10173882_

Round 1

Reviewer 1 Report

Dear Authors,

Thank you for the opportunity to review this manuscript. It is an interesting account of the assessment of adhesive capsulitis by defined MRI measures. It includes adequate details of the clinical assessment and shoulder function scores used. The MRI images were well annotated. The limitations were provided. It was generally well written and logical to follow. Thank you.

In Figures 2 and 3, is it possible to add the actual measurements obtained from these images?

Did the authors consider looking at the inter and / or intra- rater reliability of readers 1 and 2? If not available, please comment.

Minor comments:

p2 line 50 primary – primarily

p2 line 67 - From October 10, 2013, and October 16, 2017 – do you mean ‘to’ rather than ‘and’?

p2 line 92 – ‘score’ rather than ‘scored’

p4 line 192 – ‘left shoulder of the same women’ – woman.

p8 Fig 4 – capital R for Reader 2

p10 Fig 6 – please make all graphs the same y scale – 6c is different.

p11 l289-30 ‘these patients might be more suitable therapeutic options’ suitable for?

p11 l297 ‘None of MRI findings’ – none of the MRI findings

p11 l299 ‘in axillary recess’ – in the axillary recess

p11 l311 ‘the range of motion each’ – the range of motion in each

p11 l321 ‘indicator of and’ – indicator of an

Author Response

Dear Reviewer,

Thank you for the opportunity of improving our work.

Measurements have been added in the legends of the figures.

Inter-observer agreeement has been calculated, but we did not rate intra-observer agreement. This will be added in the limitations.

Figures have been updated as requested by Reviewer 2.

Text modifications have been made.

Best regards,

Reviewer 2 Report

The authors attempted to clarify the relation between MRI findings and clinical impairment of shoulder adhesive capsulitis (AC). They showed that the degree of signal intensity at the IGHL was inversely related to shoulder pain duration and range of motion, and a thickened IGHL indicated a favorable outcome at one-year follow-up.

The results are interesting for the readers and important in the diagnosis and treatment of AC.

Minor concern is the type of the graph used in the Figures 4 and 5. Line graph should be replaced with box-and-whisker plot without regression line.

Author Response

Dear Reviewer,

Thank you for the opportunity of improving our work.

Figures have been changed as aked, with a box-plot for figure 4 and barplot fot figure 5.

Best regards,

Dr R. GILLET
